# Magnetodielectric Response of Soft Magnetoactive Elastomers: Effects of Filler Concentration and Measurement Frequency

**DOI:** 10.3390/ijms20092230

**Published:** 2019-05-07

**Authors:** Sergei A. Kostrov, Mikhail Shamonin, Gennady V. Stepanov, Elena Yu. Kramarenko

**Affiliations:** 1Faculty of Physics, Lomonosov Moscow State University, Moscow 119991, Russia; kostrov@polly.phys.msu.ru (S.A.K.); gstepanov@mail.ru (G.V.S.); 2A.N. Nesmeyanov Institute of Organoelement Compounds of Russian Academy of Sciences, Moscow 119991, Russia; 3East Bavarian Centre for Intelligent Materials (EBACIM), Ostbayerische Technische Hochschule (OTH) Regensburg, Seybothstr. 2, D-93053 Regensburg, Germany; mikhail.chamonine@oth-regensburg.de; 4State Scientific Center of the Russian Federation, Institute of Chemistry and Technology of Organoelement Compounds, sh. Entuziastov 38, Moscow 111123, Russia

**Keywords:** magnetoactive elastomers, magnetodielectric effect, permittivity, dielectric loss, magnetic filler

## Abstract

The magnetodielectric response of magnetoactive elastomers (MAEs) in its dependence on filler concentration, magnetic field, and test frequency is studied experimentally. MAEs are synthesized on the basis of a silicone matrix filled with spherical carbonyl iron particles characterized by a mean diameter of 4.5 µm. The concentration of the magnetic filler within composite materials is equal to 70, 75, and 80 mass%. The effective lossless permittivity ε′ as well as the dielectric loss tanδ grow significantly when the magnetic field increases. The permittivity increases and the dielectric loss decreases with increasing filler concentration. In the measurement frequency range between 1 kHz and 200 kHz, the frequency hardly affects the values of ε′ and tanδ in the absence of a magnetic field. However, both parameters decrease considerably with the growing frequency in a constant magnetic field. The more strongly the magnetic field is applied, the larger the change in permittivity and loss tangent at the same test frequency is observed. An equivalent circuit formulation qualitatively describes the main tendencies of the magnetodielectric response.

## 1. Introduction

Magnetoactive elastomers (MAEs) are polymer-based composite materials, where micrometer-size ferromagnetic inclusions (the filler) are dispersed in a mechanically soft polymer matrix. It is well known that electric properties of MAEs significantly change in an external magnetic field [1,2,3,4,5,6,7,8]. Magnetodielectric effect (MDE) is a change of dielectric properties (effective permittivity) under an applied magnetic field. Since filler particles are electrically conductiing, the effective permittivity of the composite material is larger than the permittivity of the polymer matrix. The simplest explanation for this is that the capacitance of a parallel-plate capacitor filled by small electroconductive spheres embedded in an idealized electrical insulator is increased by a factor that depends upon the proportion of the volume occupied by the conductor, because there is no field inside the conductor [9]. Conventionally, significant MDE is observed in multiferroic materials showing magnetoelectric coupling. However, this is not the only possible way to enhance MDE. Catalan [10] has shown that strong magnetodielectric effects can also be achieved through a combination of magnetoresistance and the Maxwell–Wagner effect, which is unrelated to true magnetoelectric coupling. Similar phenomena may be expected and have been observed in MAEs, because they comprise conductive (e.g., iron) particles embedded in the electrically insulating matrix [11].

Recently, it has been shown that MDE can be further enhanced in MAEs [12,13]. In [13], we have observed the largest MDE in polymer-based composite materials at room temperature, whose order of magnitude is 10^3^%. Hitherto, there has been no generally accepted physical model that has been able to satisfactory describe such a large effect. It can be reasonably expected that the particle/matrix interfaces should play an important role in enhanced MDE in MAEs. Lunkenheimer et al. [14] thoroughly discussed the mechanisms that can lead to colossal values of the dielectric constant, particularly emphasizing effects generated by external and internal interfaces, including electronic phase separation. Psarras et al. [15] investigated polymer composites of an epoxy resin matrix with randomly dispersed iron micro-particles and revealed significant interfacial or Maxwell–Wagner–Sillars (MWS) relaxation processes for temperatures higher than 50 °C. The MWS effect is usually pronounced at very low frequencies (≤100 Hz), while significant MDE in MAEs can be observed at room temperature at frequencies higher than 1 kHz, which is typical for the characterization of real capacitors [14].

The only effect of internal interfaces on the dielectric constant seems to be insufficient for explaining large MDE in MAEs in the kHz-range. In the absence of magnetic field, the increase of the effective permittivity in comparison to the pure elastomer can be attributed to the presence of the conducting phase; the remarkable enhancement of the effective permittivity comes into play with the applied magnetic field. Tsai et al. [16] calculated that if filler particles are organized into chainlike structures rather than just randomly distributed in the elastomer matrix, the DC effective permittivity may increase by 85%, which is approximately two times smaller than the effect observed at low (1–200 kHz) frequencies in our experiments. The electric-field-induced anisotropy of particle distribution established during the polymer curing has been studied theoretically in the closely related group of electro-active polymer materials [17,18]. In comparison with isotropic MAEs, the magnetic-field-induced anisotropy of particle distribution achieved during the polymer curing is advantageous for electromagnetic shielding capability in the microwave frequency range [19].

We hypothesize that the anomalous MDE originates in the restructuring of the filler, but not necessarily due to formation of chain-like structures. In highly filled MAEs, as considered in the present paper, the formation of elongated structures may be hampered due to purely geometrical constraints [20]. Semisalova et al. [12] proposed a simple equivalent circuit that allowed one to qualitatively explain the appearance of MDE in MAEs. This model is shown in Figure 1. The dimensions of the capacitor, where one electrode is formed by an agglomerate of ideally electroconductive filler particles, change with applied magnetic field. This leads to variation of the total capacitance with magnetic field. Very recently, Isaev et al. [21] extended the approach of [12] to calculate the MAE capacitance by simulating systems of 10^5^ particles in a magnetic field, and compared the numerical results with MDE measurements at a frequency of 50 kHz. Although a qualitative agreement between simulations and experiments has been achieved, simulation results have heavily underestimated the MDE by a factor of 50.

We have now performed further experiments on several MAE samples at various test frequencies and improved this equivalent circuit. This equivalent circuit can be considered as a generalization of the MWS model combined with the Catalan’s approach. An equivalent-circuit approach has been used in the past, e.g., to explain the resistivity of carbon fiber-filled composites below the percolation threshold as a function of the volume fraction of the filler and the fiber orientation [22].

The purpose of this paper, which is a follow-up of Refs. [12,13], is twofold. First, we provide further experimental results on MDE in MAEs, where we emphasize the low-frequency and concentration dependences. Second, we formulate an equivalent circuit allowing one to qualitatively describe observed experimental dependences.

## 2. Results

MAEs containing 70, 75, and 80 mass% of iron particles were fabricated. In the following, we designate the materials as MAE70, MAE75, and MAE80, where the number means the corresponding mass concentration of iron particles. All materials have been synthesized in the absence of a magnetic field. Therefore, the spatial distribution of filler particles can be considered isotropic. Table 1 gives the overview of sample characteristics.

Figure 2 shows field dependences of the effective relative permittivity ε′ and the dielectric loss tanδ on the magnetic field for MAE samples of various compositions at three different values of the measurement frequency: f = 1 kHz, 10 kHz, and 200 kHz. Here and in the following comparison with modeling, initial curves are shown.

First of all, one can see that both the permittivity ε′ and the dielectric loss tanδ grow with increasing magnetic field for all MAE samples and all current frequencies used. The observed magnetodielectric effects, i.e., an increase in both ε′ and tanδ in magnetic fields, were previously reported in [12,13]. They were attributed to some of the restructuring of magnetic particles under the influence of magnetic field [23]. Interactions of polarized magnetic particles cause their alignment into chain-like or columnar structures along the filed direction, which can simply be observed by optical microscopy [23,24]. Enhancement of the material permittivity with initial isotropic filler distribution due to a chain-like ordering of filler particles was demonstrated by numerical modeling in [16]. It should be noted that in our samples, the filling degree is rather high, so that the magnetic particles may form percolated structures within the polymer matrix [23,24]. In this case, large-scale rearrangements of the particles with magnetic field are sterically restricted, and some anisotropy in a three-dimensional aggregate network can be expected in a magnetic field instead of a chainlike ordering of the filler.

While the maximum relative increase of the dielectric permittivity is slightly less than two times the maximum magnetic field, the growth of tanδ is more than one order of magnitude. This means that an increase in material conductivity of more than two orders of magnitude takes place, although the absolute values of it are small, being typical for dielectrics (tanδ < 1. Thus, the structural changes in filler distribution do not cause the formation of a conductive percolation network, presumably due to the presence of a dielectric layer on the surface of the iron particles that is produced in the course of surface modification. On the other hand, the distance between the magnetic particles within aggregates re-arranged in magnetic fields is expected to be smaller than the average distance between the particles in initially isotropic samples; this could facilitate hopping conductivity of the composites.

Second, one can observe a conspicuous effect of the filler concentration on both ε′ and tanδ, in particular in an applied magnetic field. In zero field, the value of permittivity grows with increasing concentration of the conducting filler; this result agrees well with conclusions of the effective medium or percolation theories [25,26,27].

The loss tangent hardly depends on the filler concentration being very low (0.005–0.02) for all the samples, i.e., the synthesized materials are good dielectrics.

In magnetic fields, the permittivity ε′ grows with the increasing concentration of magnetic particles. On the other hand, although the highest value of ε′ in the maximum magnetic field is realized for the sample with the largest filling degree (MAE80), the relative MDE—∆ε′ = (ε′(*H* = *H*_max_) − ε′(*H* = 0))/ε′(*H* = 0)—is higher for MAE75 and MAE70; it reaches 176% and 169% for these samples at *f* = 1 kHz, while it is only 154% for the MAE80 specimen. This drop of Δε′ with increasing filler content could be explained by the largest extent of the filler restructuring at intermediate filling degrees. Indeed, a compromise between enhancing magnetic interactions favoring particle restructuring and a denser packing of the particles in three-dimensional percolating filler networks together with increased rigidity of the material can be realized at some optimal particle concentration. Similar conclusion was drawn from the rheological measurements in [28]. In our case, the samples MAE70 and MAE75 do demonstrate larger magnetorheological effects than the MAE80 sample (see Table 1).

The influence of the filler content on the permittivity and dielectric loss of the material is of a contrary kind, namely, while the value of ε′ increases with increasing filler concentration, the value of tanδ decreases with it. This means that the filler concentration has a larger influence on the permittivity rather than the electrical conductivity of the composites. On the other hand, at a fixed concentration of magnetic particles, the magnetic field has a larger effect on the electrical conductivity rather than the permittivity of MAEs. A similar conclusion was drawn from the results reported in [13].

Finally, Figure 2 demonstrates the influence of the measurement frequency on the permittivity and dielectric loss of MAEs. Both ε′ and tanδ are decreasing functions of the frequency. The new result of our experimental measurements is the frequency dependence of the magnetodielectric effect. One can see in Figure 2 that the frequency dependence is considerably enhanced by a magnetic field. The larger the magnetic field, the higher the difference in both ε′ and tanδ measured at a fixed frequency. In other words, the larger the measurement frequency, the smaller the observed MDE. In the low-frequency limit (f → 0), the impedance of the equivalent circuit (see below for details) is determined by resistances of the polymer matrix and the compacted filler aggregate. In the high-frequency limit (f → ∞), the impedance of the equivalent circuit is determined by the corresponding capacitances. Magnetic field changes the arrangement of filler particles in such a way that they aggregate preferentially in the field direction, which has been experimentally verified in the past by measurements of magnetic anisotropy [29]. The resistance of the resulting aggregate grows, which is more significant at lower frequencies. Moreover, as will be shown below, the effective resistance of filler particles seems to decline with growing frequency.

In the following section, we make an attempt to rationalize the obtained results within a simple theoretical approach by giving qualitative description of the experimental data.

## 3. Discussion

Five years ago, Semisalova et al. [12] proposed a simple model that qualitatively describes MDE in MAEs. This model relies on physical intuition, and we follow this path. In the absence of an external magnetic field, the MAE sample represents a dielectric material with distributed metal particles. In the simplest case, the capacitance between the conducting plates can be estimated as that of a cylindrical metal particle (agglomerate) embedded into a polymer matrix, as shown in Figure 1a. In [12], it was assumed that the metal is an ideal electrical conductor and the capacitance of the composite was calculated as a capacitance of parallel connection of two capacitors (see Figure 1b) with different areas of plates and different distances between the plates. For the sake of simplicity, we follow this idea but assume that neither the metal is a perfect conductor nor the matrix is an ideal dielectric. The corresponding equivalent circuit is depicted in Figure 3b. The elements C_1_, R_1_, C_2_, and R_2_ refer to the polymer matrix, while C_3_ and R_3_ describe the filler. Similar approaches for conductive composite materials are mentioned in the current literature [22].

If an external magnetic field is applied to the MAE, ferromagnetic filler particles will preferably re-arrange along magnetic-field lines, and this will influence the dimensions of the equivalent cylinder, i.e., the values of all the circuit parameters. The highly conductive phase has constant volume (as the number of particles cannot change) but variable geometrical parameters (see Figure 3a). The total height of the sample is denoted d, and *x*·d is the height of the equivalent agglomerate. The diameter of the sample is 2R, and the diameter of the equivalent agglomerate is 2*y*R. Obviously, 0 < *x*,*y* < 1. The external magnetic field is directed along the axis of the cylinder. We expect elongation of the equivalent agglomerate along the magnetic field; hence, x must be a monotonously increasing function of magnetic flux density B. Additionally, the following relation must be fulfilled:(1)$$xy^{2}=p$$
where *p* is the volume fraction of the filler. If *y* = 1, there is a layer of the filler material (*x* = *p*). The expression for the effective permittivity is well known in this case, if the filler is the ideal electrical conductor [9]. In the absence of a magnetic field, the particles are homogeneously dispersed. This means that the probability to find a particle in each direction must be the same. For cylindrical symmetry, it is reasonable to assume that in the absence of magnetic field $x_{0}=y_{0}$, and
(2)$$x_{0}^{3}=p\;\Rightarrow \;x_{0}=\sqrt[{3}]{p}$$

Notice that there is a single independent parameter describing the shape of the agglomerate as we have relation (1).

Assuming the harmonic time dependence in the form of exp(*jωt*), where *j*^2^ = −1, the total admittance of the circuit shown in Figure 3b is
(3)$${\hat{Y}}_{tot}=G_{tot}+j\omega C_{tot,p}$$
where $G_{tot}$ and $C_{tot,p}$ are the total conductance and capacitance of the sample, respectively.

Circuit elements $G_{tot}$ and $C_{tot,p}$ can be easily recalculated into the effective (apparent) permittivity (*ε*′) and the effective loss tangent (tan*δ*):(4)$$\varepsilon^{\prime }=\frac{C_{tot,p}d}{\varepsilon_{0}\pi R^{2}},\;\tan\delta =\frac{G_{tot}}{\omega C_{tot,p}}$$

In (3) and (4),
(5)$$G_{tot}=\frac{1}{R_{2}}+\frac{R_{L}+\omega^{2}R_{1}R_{3}\left(T_{1}C_{1}+T_{3}C_{3}\right)}{R_{L}^{2}+\omega^{2}z^{2}}$$
(6)$$C_{tot,p}=C_{2}+\frac{R_{1}T_{1}+R_{3}T_{3}+\omega^{2}T_{1}T_{3}z}{R_{L}^{2}+\omega^{2}z^{2}}$$
where the following auxiliary parameters are used: $R_{L}=R_{1}+R_{3}$, $T_{i}=R_{i}C_{i}$, $z=R_{1}R_{3}\left(C_{1}+C_{3}\right)$, ω=2πf is the angular frequency.

Parameters $C_{1}$, $R_{1}$, $C_{2}$, and $R_{2}$ are easily expressed as functions of x and y, since they refer only to the polymer matrix.
(7)$$R_{1}=\frac{\left(1-x\right)-d}{\pi {\left(yR\right)}^{2}}\rho_{m},\;C_{1}=\frac{\pi {\left(yR\right)}^{2}}{\left(1-x\right)d}\varepsilon_{0}\varepsilon_{m}$$
(8)$$R_{2}=\frac{d}{\pi \left(1-y^{2}\right)R^{2}}\rho_{m},\;C_{2}=\frac{\pi \left(1-y^{2}\right)R^{2}}{d}\varepsilon_{0}\varepsilon_{m}$$
where $\varepsilon_{0}$ is the vacuum permittivity, $\varepsilon_{m}$ is the relative permittivity of the polymer matrix (PDMS), and $\rho_{m}$ is its electrical resistivity. The dependences of PDMS characteristics on frequency were measured in [30]; in the following, we use the values obtained in this paper, and they are given in Table 2.

The parameters $C_{3}$ and $R_{3}$ refer to the filler and they are expressed as
(9)$$R_{3}=\frac{\left(1-x\right)d}{\pi {\left(yR\right)}^{2}}\rho_{f},\;\;C_{3}=\frac{\pi {\left(yR\right)}^{2}}{xd}\varepsilon_{0}\varepsilon_{f}$$
where $\varepsilon_{f}$ and $\rho_{f}$ serve as fitting parameters. Table 2 summarizes the parameters of constitutive materials used in the modeling.

Figure 4 shows theoretical dependences of the permittivity and the loss tangent on the phenomenological parameter x of our model, which describes magnetic particle restructuring with magnetic field. For comparison, the theoretical results of the model proposed in [12] are plotted by the dash-dotted line in Figure 4a. One can see that both models predict an increase in ε′ with the increasing auxiliary parameter x, i.e., with growing magnetic field. However, the simple model of [12] (where the filler is an ideal electrical conductor) cannot describe any variation of tanδ with x; it is always the dielectric loss of the polymer matrix there. Furthermore, in contrast to the simplest model [12], the improved theoretical description can correctly reproduce the frequency dependence of both ε′ and tanδ.

Direct comparison of the theoretical dependences and the experimental results shown in Figure 4 is not possible, because the inter-relation between the phenomenological parameter x and the value of the external magnetic field is unknown. A possible solution is to exclude the unknown parameter x from consideration and compare directly the theoretical and experimental results in terms of tanδ versus ε′ values. The corresponding tanδ(ε′) curves are plotted in Figure 5 for MAEs containing various concentrations of particles and in Figure 6 for the MAE75-sample measured at various frequencies. Figure 5a and Figure 6a show the experimental results, while the model dependences are plotted in Figure 5b and Figure 6b. One can see that the theory gives us quite satisfactory agreement with the experimental results. Indeed, the model describes well an increase of the permittivity of the composites with the amount of magnetic filler, a growth of the dielectric loss with ε′, as well as a more pronounced dependence of tanδ on ε′ for lower frequencies.

As for quantitative comparison, one can see that, although the model describes properly the main tendencies in ε′ and tanδ behavior with magnetic filler concentration and frequency, it slightly underestimates the initial value of the dielectric constant of the composites, while it overestimates their tangent loss. The reason is, obviously, the limitation of the equivalent circuit representation, which does not describe the material properties in a self-consistent way like effective medium approximations [27,31] but relies on the plausibility argumentation. However, conventional effective medium theories are not capable of describing composite materials with restructuring of the filler.

## 4. Materials and Methods

### 4.1. Materials

The methods of MAE synthesis were developed by us earlier [32,33,34]. We prepared MAE samples on the basis of a two-component silicone compound commercially available under the trademark SIEL^®^ [34] filled with nominally spherical carbonyl iron microparticles (diameter of 3–5 μm). The details on the SIEL composition and MAE synthesis are available elsewhere [32,33,35]. In short, the SIEL compound consists of a vinyl-containing rubber (component A) and a hydride-containing crosslinking agent (component B). These components were mixed in a proportion A:B = 90:10 by weight to create a polymer matrix. Magnetic particles were pre-treated with silicone oil in order to enhance their surface compatibility with the silicone compound. The mixture of the polymer matrix and magnetic particles was stirred for 5 min by a mechanical mixer at room temperature, then poured into the Teflon^®^ mold and cured at 150 °C for 40 min. Disk-shaped samples with a diameter of 20.0 mm and a height of 1.1 ± 0.1 mm were cut from the fabricated MAE ribbon for the dielectric and rheological studies.

### 4.2. Methods

Rheological characterization of the MAE samples was performed using a commercial rheometer (Anton Paar, model Physica MCR 302, Graz, Austria) with a magnetorheological cell. The measurements of the shear storage modulus *G*′ and the shear loss modulus *G*″ of the MAEs were carried out in the dynamic mode of torsion oscillations in the linear viscoelastic regime at constant strain amplitude γ = 0.01% and fixed angular frequency of 10 s^−1^.

The experimental setup for dielectric measurements is sketched in Figure 7. MAE samples were placed between round electrodes forming a capacitor. The gap between electrodes was fixed during the measurements to 1.00 ± 0.01 mm, so that samples could be slightly squeezed. The diameter of electrodes was 20.0 ± 0.1 mm. The magnetic field from an electromagnet (model EM2, MAGMESS Magnet-Messtechnik Jürgen Ballanyi e.K., Bochum, Germany) was applied perpendicular to the plane of the sample. To ensure the homogeneity of an applied magnetic field, the electromagnet poles had a diameter of 10 cm, which is much larger than the capacitor’s diameter. The total height of the measurement cell was 31 mm, and the gap between electromagnet poles was 35 mm. The magnetic field was measured using the Lakeshore gaussmeter 455 DSP with the Hall sensor head HMMT-6J04-VR (Lake Shore Cryotronics, Inc., Westerville, OH, USA). The drive current I was generated by the power supply EA-PSI 8160-60 2U (EA Elektro-Automatik, Viersen, Germany). The measurements were performed in the stationary regime when the electric current was stable, i.e., the transients from switching disappeared. The magnetic field magnitude was varied between 0 and 750 mT.

An LCR meter (Hioki IM 3533-01, supplied by ASM GmbH, Moosinning, Germany) was employed to measure the parameters of the capacitor at three frequencies f = 1 kHz, 10 kHz, and 200 kHz. The measurement results were recorded in the parallel equivalent circuit mode.

The effective relative permittivity ε′ was calculated by dividing the capacitance of the capacitor filled with MAE by the capacitance of an empty capacitor with the same distance between the conducting plates.

## 5. Conclusions

We investigated the magnetodielectric response of magnetoactive elastomers based on a silicone matrix containing various concentrations of iron microparticles. The permittivity as well as dielectric loss of composites are measured in external magnetic fields up to 0.74 T at three different current frequencies of 1, 10, and 200 kHz. The following conclusions can be drawn from the analysis of the experimental data:
Both the dielectric loss tangent and the permittivity grow with the applied magnetic field. The maximum observed magnetodielectric effect reaches 179% for the sample containing 75 mass% of carbonyl iron in magnetic field of 0.74 T, while the largest absolute increase of tanδ is realized for the MAE70 sample;The permittivity grows and the dielectric loss decreases with increasing magnetic filler concentration in any magnetic field;In a fixed magnetic field, both the dielectric loss tangent and the dielectric permittivity decrease with increasing current frequency between electrodes; the stronger the magnetic field, the larger the frequency variation;There is almost no dependence of ε′ and tanδ on the current frequency in the absence of the magnetic field; however, considerable frequency dependence is induced by an applied magnetic field. The MDE and the growth of tanδ with the magnetic field are enhanced with decreasing current frequency.

The observed experimental data are rationalized within a phenomenological approach based on the equivalent circuit formulation. The main idea is that the values of the circuit elements (resistances and capacitances) change under an external magnetic field due to some restructuring of the magnetic filler described by the geometrical parameters of the highly conducting phase, which in turn are dependent on the applied magnetic field. In contrast to the previous model [12], the proposed equivalent circuit is able to catch qualitatively the main experimental observations, namely, it is sensitive to changes in measurement frequency and it can properly describe changes in conductivity of MAEs in external magnetic fields.

Our equivalent circuit representation describes the experimental results much better than the simplified model [12] or its numerical-experiment realization given in [21]. Further improvements may be achieved with the generalization of the effective medium approximation; see, e.g., [36].

## Figures and Tables

**Figure 1 ijms-20-02230-f001:**
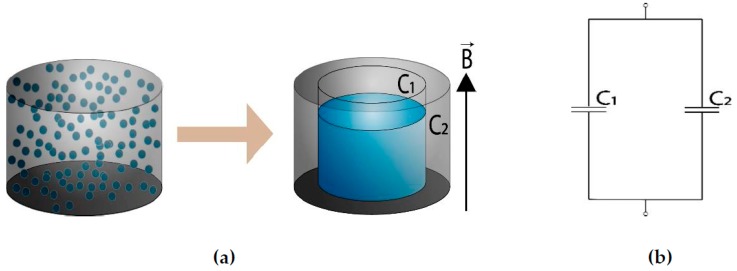
(**a**) Restructuring of the ferromagnetic filler in a magnetoactive elastomers (MAE) and (**b**) the corresponding equivalent circuit proposed in [12]. C_1_ denotes the polymer capacitance over the compacted filler, while C_2_ stands for the capacitance of the rest of the polymer matrix. The filler is assumed to be a perfect electrical conductor.

**Figure 2 ijms-20-02230-f002:**
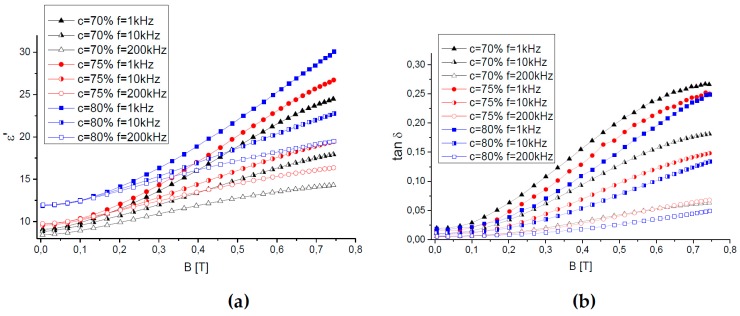
Dependence of (**a**) the relative lossless permittivity and (**b**) the dielectric loss on the magnetic field for MAEs samples measured at different frequencies of the current between electrodes.

**Figure 3 ijms-20-02230-f003:**
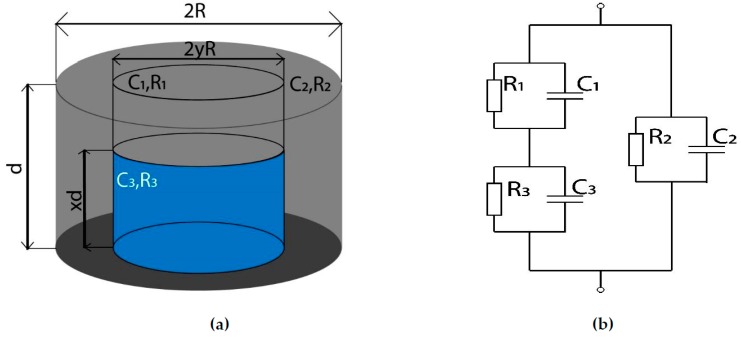
(**a**) Geometrical parameters of an MAE sample with compacted filler aggregate and (**b**) the corresponding equivalent electrical circuit.

**Figure 4 ijms-20-02230-f004:**
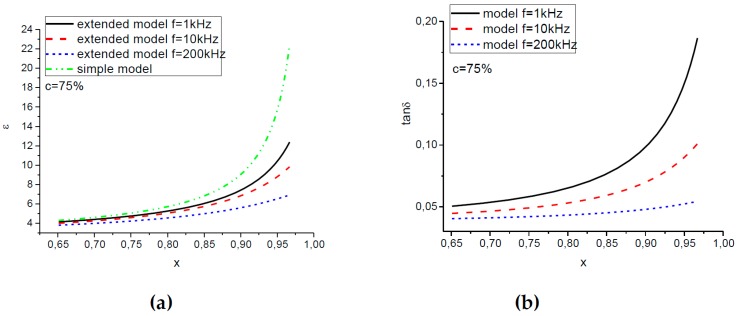
Dependence of (**a**) relative permeability and (**b**) tangent delta on geometrical parameter x. Comparison of new and old models.

**Figure 5 ijms-20-02230-f005:**
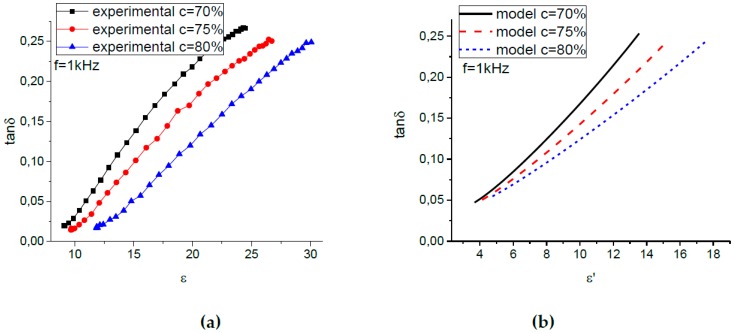
(**a**) Experimental and (**b**) model dependence of the dielectric loss tanδ on the relative permittivity ε′ for MAE samples with different iron concentrations.

**Figure 6 ijms-20-02230-f006:**
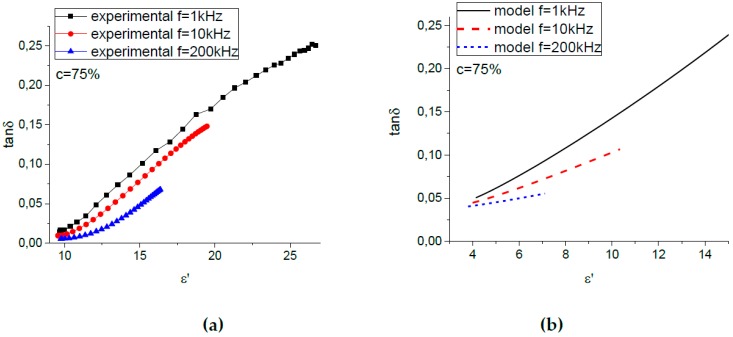
(**a**) Experimental and (**b**) model dependence of the dielectric loss tanδ on the relative permittivity ε′ for MAE75 at various measurement frequencies.

**Figure 7 ijms-20-02230-f007:**
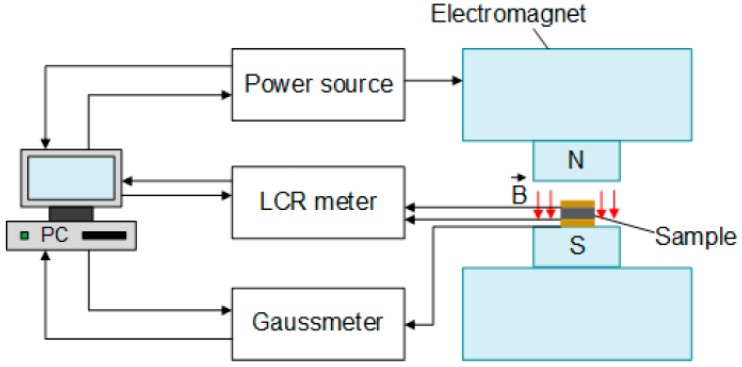
Schematic diagram of the experimental setup. Red arrows designate the applied magnetic field. Black arrows show symbolically the signal flow.

**Table 1 ijms-20-02230-t001:** Sample characteristics.

Sample	MAE70	MAE75	MAE80
Filler concentration *c*, mass%	70	75	80
Shear modulus without magnetic field *G*′_0_, kPa	28.0	29.6	33.8
Magnetorheological effect, (*G*′(*B* = 0.56T) − *G*′_0_)/*G*′_0_	33.1	31.9	17.2
Capacitance at *f* = 1 kHz, *C*_0_, pF	26.7	28.3	34.4
Magnetodielectric effect (*C*(*B* = 0.56T) − *C*_0_)/*C*_0_	1.27	1.28	1.05

**Table 2 ijms-20-02230-t002:** Specific parameters used for the model.

Test Frequency, *f*, kHz	Matrix		Filler	
	$\varepsilon_{m}$	$\rho_{m}$, Ωm	$\varepsilon_{f}$	$\rho_{f}$, Ωm
1	2.4	2 × 10^8^	69.825	6.27 × 10^5^
10	2.4	2 × 10^7^	39.27	3.14 × 10^5^
200	2.4	1 × 10^6^	22.085	6.27 × 10^4^

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
