# Peer review of "Magnetodielectric Response of Soft Magnetoactive Elastomers: Effects of Filler Concentration and Measurement Frequency"

_ijms, 2019, doi:10.3390/ijms20092230_

Reviewer 1 Report

I recommend the paper for publication in IJMS after minor revision. It is plain, well-structured and clearly presented.

There are few corrections through the text and one general comment:

- ...is depicted in Figure 4b. - should be 3b.

- formula 3 - missing index "tot" at capacitance

- fig. 3a - R2,C2 - wrong, should be R3,C3

- formula 4 - should be epsilon_eff

- there is no C_L in formulas 5 and 6, so why would you introduce it?

- i suspect it should be 10^5 instead of 105 particles

- there is no Table 2, which authors refer to. Where it is?

- did author assume frequency-independent permittivity of elastomer without magnetic particles? If yes, why? If no, what was the relation and how they made themself sure that it is not affecting the resulting data in Fig. 4?

- what are these unspecified fitting parameters ε’f and ρf in formula 9? Authors present purely theoretical calculations in fig.4, so, what was this fitting about?

- last sentence in Conclusions shall be removed.

- numeration in ref. list shall be updated - see paper 21, and all following ones.

Finally, I have to disagree with the statements like "The new result of our experimental measurements is the frequency dependence of the magnetodielectric effect" and following conclusion - since authors actually measure not MDE but simply total capacitance via admittance and capacitive reactance 1/omega*C, and then defirne "effective permittivity" using formula (4). It is not a surprise that resulted admittance will strongly depend on the used frequency. LCR meter converts the measured reactance into Farads with simple relation Xc=1/omega*C, but it does not take into account the complex structure of "capacitor" which is actually represented as R1/C2, R2/C2 and R3/C3 parts, as correctly pointed by Authors, and each of those contributes as frequency-dependent parameter. This is why the measurements of effective permittivity would rather be more precise at lower frequency (approaching quasi-static limit), which is in full agreement with observation of Authors at Fig. 4a, 4b. It is nice to present the freq-dependent measurements and describe them, but I do not think that it can be presented as ultimately new results and would recommend to re-write this part.

Also, it seems, in several times refered paper [12], there was no attempt to peform a frequency-dependent study, this is why I do not see why the conclusion should sound like "In contrast to the previous model [12], the proposed equivalent circuit is able to catch qualitatively the main experimental observations, namely, it is sensitive to changes in measurement frequency..." Authors use the same simplified approach for calculating C and extend it for broader frequency range, but there is no principal contradiction to the cited model. I would also recommend to re-phrase this part of conclusion.

Other than that, I have a positive opinion about the paper and recommend it for publishing after mentioned corrections.

Author Response

Dear Reviewer,

Please find enclosed the manuscript (Manuscript ID: ijms-495504) entitled "Magnetodielectric response of soft magnetoactive elastomers: effects of filler concentration and measurement frequency " by Sergei A. Kostrov, Mikhail Shamonin, Gennady V. Stepanov and Elena Yu. Kramarenko revised in accordance with the Reviewer comments.

Reviewer 1:I recommend the paper for publication in IJMS after minor revision.  It is plain, well-structured and clearly presented.

Response: Thank you very much for the time and effort devoted to the evaluation of our manuscript! Below are our responses to your comments.

Comment 1:There are few corrections through the text and one general comment:- ..is depicted in Figure 4b. - should be 3b.- formula 3 - missing index "tot" at capacitance- fig. 3a - R2,C2 - wrong, should be R3,C3- formula 4 - should be epsilon_eff- there is no C_L in formulas 5 and 6, so why would you introduce it?- i suspect it should be 10^5 instead of 105 particles- there is no Table 2, which authors refer to. Where it is?

Response:We are very grateful to the Reviewer for the careful reading of  the manuscript and for focusing our attention to these misprints.  They are corrected in the revised version. We are very sorry that Table 2 was deleted during final formatting of the manuscript. We have added it in the revised version. On page 6, we have explicitly specified the obtained permittivity and loss tangent as effective  coefficients.

Comment 2:- did author assume frequency-independent permittivity of elastomer  without magnetic particles? If yes, why? If no, what was the relation  and how they made themself sure that it is not affecting the resulting  data in Fig. 4?Response:The dependences of PDMS characteristics on frequency were previously  measured in [30], in our calculations we used the values obtained in  that paper, they are given in Table 2. The lossless elastomer permittivity remains practically constant in the investigated frequency range.

Comment 3:- what are these unspecified fitting parameters ε’f and ρf in formula 9?  Authors present purely theoretical calculations in fig.4, so, what was  this fitting about?Response:In the theoretical calculations, the values of the model parameters  were chosen to be close to the experimental values, in particular,  the permittivity and electrical conductivity of PDMS were taken from  the literature. However, the values of ε’f and ρf were unknown, thus, we used those which have given us the best fitting of the experimental data.

Comment 4:- last sentence in Conclusions shall be removed.Response:Done.

Comment 5:- numeration in ref. list shall be updated - see paper 21, and all  following ones.Response:Done.

Comment 6:Finally, I have to disagree with the statements like "The new result of our experimental measurements is the frequency dependence of the  magnetodielectric effect" and following conclusion - since authors actually measure not MDE but simply total capacitance via admittance  and capacitive reactance 1/omega*C, and then defirne "effective permittivity" using formula (4). It is not a surprise that resulted admittance will strongly depend on the used frequency. LCR meter converts the measured reactance into Farads with simple relation Xc=1/omega*C, but it does not take into account the complex structure of "capacitor" which is actually represented as R1/C2, R2/C2 and R3/C3 parts, as correctly pointed by Authors, and each of those contributes as frequency-dependent parameter. This is why the measurements of effective permittivity would rather be more precise at lower frequency (approaching quasi-static limit), which is in full agreement with  observation of Authors at Fig. 4a, 4b. It is nice to present the freq-dependent measurements and describe them, but I do not think that  it can be presented as ultimately new results and would recommend  to re-write this part.

Response:The employed LCR-meter can represent the measurement results both  in parallel-equivalent-circuit and series-equivalent-circuit modes. If the dielectric loss is very small, the resulting capacitances in both modes are very similar. However, if the loss tangent becomes significant, parallel and series capacitances will be significantly different as well.  For capacitors, the parallel equivalent circuit is the usual choice. The measurements of effective parameters at very low frequencies are not more precise with the employed LCR-meter, because the total impedance grows with the decreasing test frequency and will be outside of the measurement range. Moreover, the measurement time also increases with the declining test frequency and the transient material behavior becomes a very important issue. The reviewer is entirely correct that it would be also interesting to investigate the behavior of magnetodielectric effect at very low frequencies. This would require expensive equipment and we intend to do this research in the future.

We have introduced the subscript “p” (parallel) for the total capacitance Ctot,p and emphasized the parallel equivalent circuit  mode on page 9.         

Comment 7:Also, it seems, in several times refered paper [12], there was no attempt to peform a frequency-dependent study, this is why I do not see why the conclusion should sound like "In contrast to the previous  model [12], the proposed equivalent circuit is able to catch  qualitatively the main experimental observations, namely,  it is sensitive to changes in measurement frequency..." Authors use the same simplified approach for calculating C and extend it for  broader frequency range, but there is no principal contradiction to the cited model. I would also recommend to re-phrase this part of conclusion.

Response: The reviewer is entirely correct: in the pioneering work [12], only the dependence of the permittivity on applied magnetic field at  a fixed test frequency has been measured. However, the point is that  the original equivalent circuit of [12] (cf. Fig. 1) is not capable of  describing experimentally observed dependences of dielectric loss on  test frequency and fill factor. In fact, the dielectric loss on the old  model [12] is always the dielectric loss of the polymer matrix.

We have introduced several changes on pages 2 and 7 according to the reviewer’s comment.

Other than that, I have a positive opinion about the paper and recommend it for publishing after mentioned corrections.Response: Thank you very much for a positive general evaluation of our work!

All the changes to the manuscript made according to the reviewers comments are highlighted by yellow color in the revised version of the manuscript.

Finally, we would like to thank the Reviewers whose valuable comments helped us

a lot in improving the manuscript.

Reviewer 2 Report

Magneto-active elastomers (MAE) are nowadays promising candidates for field-responsive functional materials. Soft robotics is an emerging field of research where MAE is extensively used as smart materials. Hence topics of the manuscript is timely and relevant. Moreover, the issue is presented in a nice manner with a good standard of english writing. Hence, I would recommend this manuscript to publish in IJMS after a revision. In this case, following modifications need to considered

---Equation 4(a) needs to be corrected. What is '?' sign in the equation?

----Please be consistent in writing equations (better to use LATEX). For example,

ε_f is  written in Eqn 9 but ε'_f is written in line 247. Similarly,   ρ (e.g. ρ_f)  in equations (7-9) are not consistent

---For a better visualization purpose, use different legends (dot, dot-dash, dash-dash etc) in Fig 4, similar to Fg 2

---I do not understand the lines 269-270 ....' one can see that the theory gives us----'

However, authors did not compare directly experiments with models (see Fig 6a and 6b). I would recommend to merge Figs 6a and 6b in one figure so that readers can see how models match with experiments

---In Lines 123-124, authors talk about chain-like ordering of particles and related material modelling citing Tsai et al. (2017). I would recommend more related modelling references here. In this case, authors should put these references--

-M. Hossain and P. Steinmann (2018), Modelling electro-active polymers with a dispersion-type anisotropy, Smart Materials and Structures, 27(2):1-24

M. Hossain (2019), Modelling the curing process in particle-filled electro-active polymers with a dispersion anisotropy, Continuum Mechanics and Thermodynamics, DOI:10.1007/s00161-019-00747-5

The reviewer reads the manuscript with a great interest and look forward a revised version.

Author Response

Dear Reviewer,

Please find enclosed the manuscript (Manuscript ID: ijms-495504) entitled "Magnetodielectric response of soft magnetoactive elastomers: effects of filler concentration and measurement frequency " by Sergei A. Kostrov, Mikhail Shamonin, Gennady V. Stepanov and Elena Yu. Kramarenko revised in accordance with the Reviewer comments.

Reviewer 2Magneto-active elastomers (MAE) are nowadays promising candidates for field-responsive functional materials. Soft robotics is an emerging field of research where MAE is extensively used as smart materials. Hence topics of the manuscript is timely and relevant. Moreover, the issue is presented in a nice manner with a good standard of english writing. Hence, I would recommend this manuscript to publish in IJMS after a revision. In this case, following modifications need to considered

Response: Thank you very much for the time and effort devoted to the evaluation of our manuscript! We greatly appreciate your positive general evaluation and constructive suggestions. Below are our responses to your comments.

Comment 1:---Equation 4(a) needs to be corrected. What is '?' sign in the equation?-Please be consistent in writing equations (better to use LATEX). For example,ε_f is written in Eqn 9 but ε'_f is written in line 247. Similarly, ρ (e.g. ρ_f) in equations (7-9) are not consistentResponse:We are very grateful to the Reviewer for pointing to some misprints and formatting errors. We have corrected them.

Comment 2:---For a better visualization purpose, use different legends (dot, dot-dash, dash-dash etc) in Fig 4, similar to Fg 2Response:Done.

Comment 3:---I do not understand the lines 269-270 ....' one can see that the theory gives us----'However, authors did not compare directly experiments with models (see Fig 6a and 6b). I would recommend to merge Figs 6a and 6b in one figure so that readers can see how models match with experimentsResponse: We do not pretend that our simplified model gives a quantitative agreement with the experimental results. Merging Figures 6a and 6b into a single Figure makes it hardly understandable, because the experimental and theoretical curves will cross each other. By using two separate Figures we emphasize the qualitative similarities and the correct order of magnitude of calculated parameters.

The passage starting at lines 269-270 has been modified according to the reviewer’s comment.

Comment 4:---In Lines 123-124, authors talk about chain-like ordering of particles and related material modelling citing Tsai et al. (2017). I would recommend more related modelling references here. In this case, authors should put these references---M. Hossain and P. Steinmann (2018), Modelling electro-active polymers with a dispersion-type anisotropy, Smart Materials and Structures, 27(2):1-24M. Hossain (2019), Modelling the curing process in particle-filled electro-active polymers with a dispersion anisotropy, Continuum Mechanics and Thermodynamics, DOI:10.1007/s00161-019-00747-5Response: Done.

The reviewer reads the manuscript with a great interest and look forward a revised version

Response:We are very grateful to the Reviewer for the high estimate of our work.

All the changes to the manuscript made according to the reviewers comments are highlighted by yellow color in the revised version of the manuscript.

Finally, we would like to thank the Reviewers whose valuable comments helped us a lot in improving the manuscript.

Reviewer 3 Report

The manuscript submitted by Kramarenko et al. presents the evaluation of magnetodielectric response of magnetorheological elastomers in dependence on magnetic particles concentration, magnetic field applied and test frequency. This study extend the already done work in this area by giving the response in broader range of frequencies and also formulate an equivalent circuit for qualitative description of obtained dependences which could be interesting from practical viewpoint and, thus, it can be published in International Journal of Molecular Sciences. However, there are some points within the manuscript to be answered before its publication:

1. I am not sure whether the structure of the manuscript is given by the publisher but it could be quite difficult for the reader to start directly with the results and discussion of the obtained results without the knowledge of materials involved in magnetorheological elastomer investigated and the methodology of experiments.

2. The introduction into magnetorheological elastomers is to the point and almost comprehensive. However, some recent study dealing with the possible applications of magnetodielectric response of magnetorheological elastomers in shielding applications could be also interesting for the readership. Please, see: “Smart Materials and Structures, 2017, vol. 26, issue 9, art. no. 095005.”

3. Regarding the Figure 2. The effect of magnetic field on the magnetodielectric response is discussed within the corresponding text in the manuscript, but this is not in the mentioned figure (it seems it is just without the magnetic field applied). What intensity of magnetic field was applied in this experiment?

4. Did the proposed theory take into account also the material parameters of the elastomeric matrix (hardness, Poisson's ratio of filled elastomer composite etc.)?

5. The inaccuracies in Figures numbering (Figure 4b versus 3b at the beginning of the Discussion section), missing Table 2 or symbols used in equations should be checked.

Author Response

Dear Reviewer,

Please find enclosed the manuscript (Manuscript ID: ijms-495504) entitled "Magnetodielectric response of soft magnetoactive elastomers: effects of filler concentration and measurement frequency " by Sergei A. Kostrov, Mikhail Shamonin, Gennady V. Stepanov and Elena Yu. Kramarenko revised in accordance with the Reviewer comments.

Reviewer 3The manuscript submitted by Kramarenko et al. presents the evaluation of magnetodielectric response of magnetorheological elastomers in dependence on magnetic particles concentration, magnetic field applied and test frequency. This study extend the already done work in this area by giving the response in broader range of frequencies and also formulate an equivalent circuit for qualitative description of obtained dependences which could be interesting from practical viewpoint and, thus, it can be published in International Journal of Molecular Sciences. However, there are some points within the manuscript to be answered before its publication:

Response: Thank you very much for the time and effort devoted to the evaluation of our manuscript! We greatly appreciate your positive general evaluation and constructive suggestions. Below are our responses to your comments.

Comment 1.I am not sure whether the structure of the manuscript is given by the publisher but it could be quite difficult for the reader to start directly with the results and discussion of the obtained results without the knowledge of materials involved in magnetorheological elastomer investigated and the methodology of experiments.

Response: Yes, the structure of the manuscript is prescribed by the publisher. That is why we placed the Table 1 in the beginning of the Results section. It immediately provides the summary of the relevant properties of investigated materials.

Comment 2.The introduction into magnetorheological elastomers is to the point and almost comprehensive. However, some recent study dealing with the possible applications of magnetodielectric response of magnetorheological elastomers in shielding applications could be also interesting for the readership. Please, see: “Smart Materials and Structures, 2017, vol. 26, issue 9, art. no. 095005.”

Response: Done.

Comment 3.Regarding the Figure 2. The effect of magnetic field on the magnetodielectric response is discussed within the corresponding text in the manuscript, but this is not in the mentioned figure (it seems it is just without the magnetic field applied). What intensity of magnetic field was applied in this experiment?

Response: The value of the applied magnetic flux density is provided on the horizontal axis. No magnetic field has been applied during synthesis. Corresponding sentences have been inserted on page 3.

Comment 4.Did the proposed theory take into account also the material parameters of the elastomeric matrix (hardness, Poisson's ratio of filled elastomer composite etc.)?

Response: The proposed simplified model is not capable of predicting the dependencies of dielectric parameters on the material properties of the elastomeric matrix, because it is not clear how the auxiliary parameter x will depend on them. This is one of the reasons why we eliminated it in the representation of obtained results.

Comment 5.The inaccuracies in Figures numbering (Figure 4b versus 3b at the beginning of the Discussion section), missing Table 2 or symbols used in equations should be checked

Response: We are very grateful to the Reviewer for pointing to some misprints and formatting errors. We have corrected them.

All the changes to the manuscript made according to the reviewers comments are highlighted by yellow color in the revised version of the manuscript.

Finally, we would like to thank the Reviewers whose valuable comments helped us a lot in improving the manuscript.

Round  2

Reviewer 2 Report

It can be accepted